

# Examination of reliability and validity of the Self-Assessment Burden Scale-Motor for community-dwelling older adults in Japan: a validation study

Hiroshi Warabino[1,2], Toshikatsu Kaneda[3], Yuma Nagata[4], Katsushi Yokoi[4], Kazuyo Nakaoka[4], Yasuhiro Higashi[3], Yoshimi Yuri[3], Hiroko Hashimoto[3] and Shinichi Takabatake[5]

[1] Graduate School of Comprehensive Rehabilitation, Osaka Prefecture University, Habikino, Osaka, Japan
[2] Medicare-Rehabili Home-Visit Nursing Station, Osaka, Japan
[3] Faculty of Rehabilitation, Morinomiya University of Medical Sciences, Osaka, Japan
[4] Graduate School of Rehabilitation Science, Osaka Metropolitan University, Habikino, Osaka, Japan
[5] Faculty of Health Sciences, Kyoto Tachibana University, Kyoto, Japan

Corresponding author
Hiroshi Warabino,
hirosi2130@gmail.com

## ABSTRACT

**Background**. The aging society in Japan is progressing rapidly compared with that in the United States and European countries. Aging limits activities of daily living (ADL) in older adults, declining their lives and functions at home. Therefore, improving their ADL to effectively support their functioning at home for as long as possible is vital. Consequently, supporters need to have a common understanding, be promptly aware of the decline in ADL, and quickly introduce rehabilitation. The Functional Independence Measure (FIM) and Barthel Index (BI) are the main scales used to quantitatively assess ADL. However, previous studies have reported that FIM requires specialized knowledge for evaluation, and BI does not appropriately capture changes in ADL. The Self-Assessment Burden Scale-Motor (SAB-M) was developed as a scale for family caregivers to appropriately assess changes in ADL in older adults. Previous studies using the SAB-M have confirmed its reliability and validity in hospitalized patients as assessed by their family caregivers. Therefore, this study aimed to investigate the reliability and validity of the SAB-M among community-dwelling older adults as assessed by their family caregivers.

**Methods**. This study included community-dwelling older adults who received home-visit rehabilitation at the first author's facility between October 2020 and December 2020 in Japan. Following previous studies, the SAB-M was used by family caregivers to assess 20 older adults twice for intra-rater reliability. Furthermore, 168 older adults were evaluated by family caregivers for internal consistency using the SAB-M. For criterion validity, the SAB-M was used for the assessment by family caregivers, and therapists used the FIM-Motor (FIM-M). This study used the weighted kappa, Cronbach's alpha, and Spearman's rank correlation coefficients for the statistical analysis of intra-rater reliability, internal consistency, and criterion validity, respectively.

**Results**. The weighted kappa coefficient for the total score was 0.98 ($p < 0.01$) and individual item, it was 0.93 for feeding ($p < 0.01$), 0.91 for bathing ($p < 0.01$), 0.98 for dressing ($p < 0.01$), 0.94 for transfer ($p < 0.01$), 0.94 for walking/wheelchair ($p < 0.01$), 0.95 for stairs ($p < 0.01$), and 0.96 for bladder management ($p < 0.01$). The Cronbach's

alpha was 0.93 for the seven items. The Spearman's rank correlation coefficient between the SAB-M and FIM-M scores was 0.91 ($p < 0.01$).

**Conclusion**. The SAB-M has sufficient reliability and validity among community-dwelling older adults. Family caregivers can routinely assess changes in the ADL of community-dwelling older adults using the SAB-M, enabling them to promptly consider introducing rehabilitation when older adults' ADL declines. Therefore, implementing SAB-M helps older adults live and function at home for as long as possible.

# INTRODUCTION

Rapid aging is a global trend, with projections indicating an increase in the population aged >60 years to 1.4 billion by 2030 and further escalating to 2.1 billion by 2050 (*World Health Organization, 2022*). Notably, 28.6% of the population in Japan was aged ≥65 years in 2020, highlighting a more accelerated progression towards an aging society compared with the United States and European nations (*Statistical Handbook of Japan, 2022*).

Activities of daily living (ADL) are crucial for maintaining a high-quality home life in older adults. As individuals age, there is an observable decline in ADL due to cognitive and functional impairments (*Suzuki, 2018*). A 7-year longitudinal study of community-dwelling older adults in the Netherlands identified ADL impairment as a significant predictor of mortality (*Gobbens & Ploeg, 2020*). Systematic reviews have highlighted that the primary predictors for the institutionalization of older adults, especially in nursing homes, correlate strongly with a decline in ADL and insufficient supportive assistance (*Luppa et al., 2010*). Furthermore, decreased ADL is associated with a deterioration of the quality of life (QOL) among older adults (*Yaya et al., 2020*). Therefore, prioritizing ADL in the care and support of older adults is essential to enable them to live independently at home.

The introduction of rehabilitation is effective for community-dwelling older adults when ADL declines, and caregivers need to be promptly aware of their disability. Research showed that approximately 30% of frail older adults in the community are impaired in at least one ADL (*Wong et al., 2010*). Furthermore, systematic reviews have demonstrated that rehabilitation interventions are effective for community-dwelling older adults who face ADL challenges (*Liu, Chang & Chang, 2018*). In addition, functional interventions such as resistance exercise and balance exercise positively impact ADL and QOL and reduce the fear of falling among older adults (*Akosile et al., 2021*; *Campbell et al., 2021*). However, high-need/high-cost older adults experience negative effects due to a lack of access to professional care (*Beach et al., 2020*). With the rising number of care-dependent older adults, the prevalence of informal caregiving, primarily offered by family members, is also increasing (*Broese van Groenou & De Boer, 2016*). Consequently, we posit that there is a high likelihood of inadequate collaboration between professionals and informal caregivers.

If the daily evaluation of ADL by caregivers is conducted correctly, prompt access to professional and initiation of rehabilitative interventions may help older adults sustain their residential independence.

The Functional Independence Measure (FIM) (*Mcdowell, 1997*) and Barthel Index (BI) (*Mahoney & Barthel, 1965*; *Schlote et al., 2004*) are commonly used to quantitatively assess ADL. However, FIM requires specialized knowledge for evaluation, and it is necessary to implement considerable human and time costs (*Yamada et al., 2006a*; *Yamada et al., 2006b*). Previous studies have also indicated that BI can be inaccurate in capturing changes in ADL (*Morton, Keating & Davidson, 2008*). In Japan, various instruments have been developed to assess the ADL in community-dwelling older adults, such as the questionnaire version of the FIM (*Ota et al., 1997*), the short version of the FIM (*Yamada et al., 2006a*), iFIM (*Yamada et al., 2006b*), and the University of Occupational and Environmental Health (UOEH) self-assessment version of BI (*Fukuda, Takemoto & Shirayama, 1995*). However, their complexity, large number of questions, and the amount of time required to answer them render these tools impractical for use. Therefore, for the effective daily assessment of ADL in community settings, it is essential that the scale is easily accessible to caregivers without specialized knowledge and can quantitatively and appropriately assess changes in ADL. Consequently, the Self-Assessment Burden Scale-Motor (SAB-M) was developed to address this need (*Kaneda et al., 2020a*). The SAB-M consists of seven items, including feeding, bathing, dressing of the lower body, bed/chair/wheelchair transfer, walking/wheelchair, stairs, and bladder management. Caregivers rate the patient's need for assistance on a four-point ordinal scale ranging from 1 (full assistance required) to 4 (no assistance needed) (See "Instrumentation" section for details). Previous studies employing the SAB-M have confirmed its reliability and validity in hospitalized patients, as assessed by family caregivers (*Kaneda et al., 2020b*). However, the reliability and validity of the SAB-M in community-dwelling older adults are yet to be thoroughly examined.

This study aimed to confirm the reliability and validity of the SAB-M among community-dwelling older adults as assessed by their family caregivers.

## MATERIALS & METHODS

### Study design

This study examined the reliability and validity of the SAB-M.

We examined test-retest reliability and internal consistency for reliability, and criterion validity for validity.

### Setting

This study included community-dwelling older adults who received home-visit rehabilitation from the first author's facility between October 2020 and December 2020 in Japan. Data collection was conducted at the participants' homes. In Japan, home-visit rehabilitation is the rehabilitation provided by a physical therapist or occupational therapist in the participants' home. The focus is not only on the physical function but also on basic ADL, instrumental ADL, and QOL. Therefore, rehabilitation also focuses on promoting

activity, encouraging social participation, and enabling independent home living (*Asano et al., 2019*).

## Participants

Older adults were selected as study participants based on the following inclusion criteria: (a) living at home for at least 3 months, because the recovery rate has been reported to be more stable starting from 3 months after discharge (*Li et al., 2020*), (b) being cared for by healthy family caregivers, and (c) receiving home rehabilitation services. The exclusion criteria included being cared for by family caregivers with dependent ADL and dementia. The first author randomly selected participants who met the above criteria.

We aimed to recruit 100 participants for the test-retest reliability group and 200 participants for the internal consistency and criterion validity group, according to the COSMIN checklist (*Mokkink et al., 2019*) and previous study (*Kaneda et al., 2020b*).

## Variables

We obtained the following information from medical records: age, sex, and diagnosis (for older adults), and age, sex, and relationship with older adults (for family caregivers).

We used SAB-M scores for test-retest reliability and internal consistency, and SAB-M scores and FIM-Motor (FIM-M) scores for criterion validity.

## Instrumentation
### *Self-Assessment Burden Scale-Motor (SAB-M)*

The SAB-M is an ADL evaluation questionnaire based on several ADL evaluation instruments, such as the Katz Index of Independence in daily living (*Katz, Hedrick & Henderson, 1979*), Physical Self-Maintenance Scale (*Lawton & Brody, 1969*), Rapid Disability Rating Scale (*Linn & Linn, 1982*), UOEH self-assessment version of the BI (*Fukuda, Takemoto & Shirayama, 1995*), and the short version of the FIM (*Yamada et al., 2006a*). Four occupational therapists with at least 5 years of clinical experience conducted six review sessions to examine content validity. The assessment items, question content, and wording of the questions were also examined. Caregivers completed the SAB-M based on their observations of older adults. The SAB-M can be used for mail surveys and can be evaluated in approximately 5 min. The SAB-M comprises seven items (Feeding, Bathing, Dressing the lower body, bed/chair/wheelchair transfer, Walk/Wheelchair, Stairs, Bladder management), and each item is scored on a four-point Likert scale.

Caregivers rated the older adults' need for assistance rather than their degree of independence. The items included "feeding", which refers to carrying the food to the mouth; "bathing", which refers to washing the body under the head; "dressing the lower body", which refers to putting pants on; and "bed/chair/wheelchair transfer", which refers to moving oneself from the bed to a chair or wheelchair. For the "walk/wheelchair" item, "walk" or "wheelchair" should be chosen based on the older adult's primary way of moving in daily life. "Stairs" refers to going up and down the stairs, and "Bladder management" refers to urination failure and the amount of assistance required to urinate. Each item was scored using an ordinal scale from 1 to 4 (1: full assistance needed; 2: physical care needed; 3: takes time, supervision, or environmental support needed; and 4: no assistance

needed). The SAB-M consists of eight pages because it includes figures and tables designed to facilitate the rating process for caregivers. The Rasch analysis was used to verify the structural validity and reliability of the SAB-M prototype. Georg Rasch developed this analysis in the 1960s (*Bond & Fox, 2015*), and it is a Modern Test Theory. It is often used in rehabilitation (*Árnadóttir, Löfgren & Fisher, 2010*). The Rasch analysis revealed that the four-item and seven-item scales were appropriate (*Kaneda et al., 2020a*). Finally, we examined the intra-rater reliability and criterion validity in inpatient settings using a Classical Test Theory. For the intra-rater reliability analysis, the weighted kappa coefficient for the seven items was >0.75. For criterion validity analysis, Spearman's correlation coefficient was 0.85 for the total score between the SAB-M and FIM-M (*Kaneda et al., 2020b*).

The authors had the permission to use this instrument from the copyright holders.

## Functional independence measure-motor

Developed by Granger in 1983, the FIM is an ADL assessment method measuring a person's level of disability in terms of the burden of care. The FIM has a motor domain (FIM-M) and a cognitive domain (FIM-C); FIM-M was used in this study. The FIM-M comprises 13 items based on 4 domains (self-care, sphincter control, transfers, locomotion). Each FIM-M item is scored on a 7-point ordinal scale ranging from total assistance (or complete dependence) to complete independence. FIM-M is considered as the gold standard for ADL assessment because it can capture objective and detailed changes.

## Procedure and data analysis
### Test-retest reliability

Family caregivers evaluated the older adults' ADL using the SAB-M. They conducted two evaluations separated by a 2-week interval. This period between the test and retest was selected because it was considered sufficiently long to prevent recall of the previous answers but short enough to not allow for significant changes in the participants' conditions. The total score and agreement of each item between the first and second SAB-M scores were analyzed using weighted kappa coefficients. This study's interpretation of the agreement was based on the criteria of *Landis & Koch (1997)* (coefficient <0: poor agreement, coefficient 0–0.2: slight agreement, coefficient 0.21–0.40: fair agreement, coefficient 0.41–0.60: moderate agreement, coefficient 0.61–0.80: substantial agreement, coefficient 0.80–1.0: almost perfect agreement).

## Internal consistency

Family caregivers evaluated the ADL of older adults using the SAB-M, and Cronbach's alpha was calculated based on the item scores of all participants.

## Criterion validity

Family caregivers evaluated the ADL of the older adults using the SAB-M, whereas therapists used the FIM-M. Family caregivers and therapists did not know each other's evaluation results. The correlation between the family caregivers' SAB-M total score and the therapists' FIM-M total score was analyzed using Spearman's rank correlation coefficient.

The correlation was interpreted based on Evans' criteria (*Evans, 1996*), where a correlation of <0.2 was considered very weak, 0.2 to <0.4 was considered weak, 0.4 to <0.6 was considered moderate, 0.6 to <0.8 was considered strong, and a correlation of ≥0.8 was considered very strong.

Statistical analyses were performed using SPSS version 25, and a list-wise deletion method was used for missing data.

### Ethical considerations

All older adults and their families provided written informed consent before participating in this study. The Graduate School of Comprehensive Rehabilitation, Osaka Prefecture University's research ethics committee approved this study (2019–215). All procedures adhered to the latest version of the Declaration of Helsinki guidelines.

## RESULTS

### Participants

We recruited 200 participants who met our inclusion criteria. After being informed about the study, 16 individuals chose not to participate. In addition, 8.7% of the participants had missing data, resulting in a dataset of 168 individuals. Furthermore, 20 participants were randomly selected for the test-retest reliability. In total, 168 participants were evaluated for internal consistency and criterion validity.

### Test-retest reliability

The test-retest group included 20 participants (average age = 80.2 years, standard deviation (SD) = 12.3, range 51–95) as shown in Table 1. This table details their characteristics, including FIM-M and SAB-M information. In addition, 20 family caregivers (average age = 62.5 years, SD = 10.6, range 45–84) were involved in this study to assess the SAB-M. The weighted kappa coefficient for the total score was 0.98 ($p < 0.01$). The individual item coefficients were as follows: 0.93 for feeding ($p < 0.01$); 0.91 for bathing ($p < 0.01$); 0.98 for dressing ($p < 0.01$); 0.94 for transfer ($p < 0.01$); 0.94 for walking/wheelchair ($p < 0.01$); 0.95 for stairs ($p < 0.01$); and 0.96 for bladder management ($p < 0.01$), as shown in Table 2. The overall score and individual item agreement suggested a near-perfect test-retest reliability.

### Internal consistency

The study involved 168 participants (average age = 78.1 years, SD = 11.5, range 40–100), as shown in Table 3. This table details their characteristics, including FIM-M and SAB-M scores. In addition, 168 family caregivers (average age = 68.4 years, SD = 11.1, range 34–93) were involved in this study to assess the SAB-M. The Cronbach's alpha coefficient for the seven items was 0.93.

### Criterion validity

The participants and family caregivers were the same as those in the internal consistency study (Table 3). In addition, 12 therapists (average age of experience = 9.7 years, SD = 3.2, range 4–15) were included in this study to assess the FIM-M. The correlation coefficient between SAB-M and FIM-M total scores was 0.91 ($p < 0.01$), indicating a strong correlation.

**Table 1 Demographic information of participants and family caregivers in test-retest reliability.**

| Participants (*n* = 20) | | Family caregivers (*n* = 20) | |
|---|---|---|---|
| Age | | Age | |
| Mean | 80.2 | Mean | 62.5 |
| SD | 12.3 | SD | 10.6 |
| Range | 51–95 | Range | 45–84 |
| Sex | | Sex | |
| Male | 8 | Male | 5 |
| Female | 12 | Female | 15 |
| Diagnosis | | Relationship | |
| Cerebral vascular accident | 8 | Wife | 4 |
| Orthopedic disease | 3 | Daughter | 11 |
| Intractable disease | 1 | Husband | 4 |
| Internal disorder | 3 | Son | 1 |
| Dementia | 1 | | |
| Disuse syndrome | 4 | | |
| FIM-M | | | |
| Mean | 63.4 | | |
| SD | 23.5 | | |
| SAB-M | | | |
| Mean | 18.2 | | |
| SD | 6.6 | | |

Notes.
FIM-M, Functional Independence Measure-Motor; SAB-M, Self-Assessment Burden Scale-Motor.

**Table 2 Weighted kappa coefficient of Self-Assessment Burden Scale-Motor.**

| SAB-M | Weighted kappa coefficient (*n* = 20) |
|---|---|
| Total score | 0.98[*] |
| Feeding | 0.93[*] |
| Bathing | 0.91[*] |
| Dressing | 0.98[*] |
| Transfer | 0.94[*] |
| Walking/Wheelchair | 0.94[*] |
| Stairs | 0.95[*] |

Notes.
SAB-M, Self-Assessment Burden Scale-Motor.
[*]$p < 0.01$

# DISCUSSION

This study examined the reliability and validity of the SAB-M as a tool for family caregivers to assess the ADL of community-dwelling older adults. Our findings suggest that the SAB-M has sufficient reliability and validity among community-dwelling older adults. Family caregivers can routinely assess changes in the ADL of community-dwelling older adults using the SAB-M, enabling them to promptly consider introducing rehabilitation

**Table 3** Demographic information of participants and family caregivers in Internal Consistency and Criterion validity.

| Participants ($n = 168$) | | Family caregivers ($n = 168$) | |
|---|---|---|---|
| Age | | Age | |
| Mean | 78.1 | Mean | 68.4 |
| SD | 11.5 | SD | 11.1 |
| Range | 40–100 | Range | 34–93 |
| Sex | | Sex | |
| Male | 74 | Male | 52 |
| Female | 94 | Female | 116 |
| Diagnosis | | Relationship | |
| Cerebral vascular accident | 59 | Wife | 57 |
| Orthopedic disease | 38 | Daughter | 50 |
| Intractable disease | 28 | Husband | 40 |
| Internal disorder | 26 | Son | 11 |
| Dementia | 9 | Sibling | 4 |
| Disuse syndrome | 8 | Parent | 6 |
| FIM-M | | | |
| Mean | 62.9 | | |
| SD | 24.0 | | |
| SAB-M | | | |
| Mean | 17.6 | | |
| SD | 6.5 | | |

**Notes.**

FIM-M, Functional Independence Measure-Motor; SAB-M, Self-Assessment Burden Scale-Motor.

when older adults' ADL declines. Therefore, we believe that implementing the SAB-M will help older adults continue living at home for as long as possible.

Overall, 8.7% of the data were missing, caused by skipping pages due to the multiple-page format of the SAB-M. However, in another study that employed the same questionnaire for family caregivers as the one in the current study, the missing data rate was 18% (*Ris et al., 2022*), whereas in this study it was lower than this value. This was attributed to the minimal number of questions and ease of answering them. *Rodgers & Miller (1997)* highlighted the importance of using unambiguous wording in their comparative analysis of ADL questions in older adult surveys. The SAB-M survey asked caregivers to rate the degree of assistance they required, which may have made it easier for them to provide accurate responses. This clear wording may have contributed to the survey's low rate of missing data, since caregivers were more likely to understand and answer the questions accurately. The low rate of missing data in the SAB-M suggests it is a readily evaluable measure. To better address the issue of missing data in the future, we are considering adding a checklist at the end of the document as a solution.

The test-retest reliability demonstrated a high level of agreement with nearly perfect results for the total score and score for each item. This indicates that the SAB-M is a reproducible assessment tool. Cronbach's alpha was 0.93, indicating an almost perfect internal consistency. The SAB-M measures the ADL of community-dwelling older adults

as a whole. Criterion validity was highly significant, with strong correlation coefficients. This indicates that the SAB-M is a valid method for assessing ADL in community-dwelling older adults. In this study, the levels of reliability and validity were higher than those reported in previous studies for all the items. In a previous study, the SAB-M was evaluated by family caregivers who predicted the assistance required for ADL after hospital discharge (*Kaneda et al., 2020b*). This assessment was performed because the patients were about to be discharged. However, this study evaluated the SAB-M based on the assistance provided for ADL in a community setting over a specified period. The higher values found in this study compared with those in previous studies may be because the SAB-M was evaluated based on stable assistance provided for ADL in a community setting over a specified period rather than on predicted ADL after hospital discharge. Caregivers can use the SAB-M to regularly assess the ADL in older adults without professional involvement. If a decline in ADL is observed, early intervention by specialists, such as rehabilitation, is recommended. We believe that the SAB-M facilitates smooth access between caregivers and professionals.

## LIMITATIONS

This study has some limitations. The sample size of 20 participants for assessing the test-retest reliability was small. The sample size of 20 is considered to provide a degree of reliability based on previous research. According to the COSMIN checklist (*Mokkink et al., 2019*), a sample size of >100 participants was considered very good, 50–99 participants were adequate, 30–49 participants were doubtful, and ≤30 participants were inadequate. Therefore, this study's sample size of 20 participants was insufficient, and future studies should aim to increase the sample size to enhance the reliability of the findings. Additionally, only family caregivers evaluated the ADL of the participants. Given the increasing prevalence of caregiving situations, such as living alone or being cared for by elderly caregivers as the population declines, it is important to determine whether the SAB-M is useful for professional caregivers. Furthermore, the participants in this study were community-dwelling older adults who received home-visit rehabilitation. In order to generalize the results of the study, it is necessary to include community-dwelling older adults who do not receive home-visit rehabilitation as participants.

## CONCLUSIONS

The SAB-M has sufficient reliability and validity among community-dwelling older adults. If caregivers routinely assess the ADL of community-dwelling older adults using the SAB-M, they can promptly consider introducing rehabilitation. Early detection and support of a decline in ADL can help older adults continue living at home for as long as possible and maintain a better QOL.

In the future, we would like to increase the number of participants. In addition, we plan to determine whether the SAB-M is useful for professional caregivers and to include community-dwelling older adults who do not receive home-visit rehabilitation as participants. We believe this approach will make it a more useful evaluation tool.

## ACKNOWLEDGEMENTS

We thank the participants and their families for their valuable contributions to this study.

### Funding

The authors received no funding for this work.

### Competing Interests

The authors declare there are no competing interests.

### Author Contributions

- Hiroshi Warabino conceived and designed the experiments, performed the experiments, analyzed the data, prepared figures and/or tables, authored or reviewed drafts of the article, and approved the final draft.
- Toshikatsu Kaneda conceived and designed the experiments, analyzed the data, authored or reviewed drafts of the article, and approved the final draft.
- Yuma Nagata conceived and designed the experiments, authored or reviewed drafts of the article, and approved the final draft.
- Katsushi Yokoi conceived and designed the experiments, authored or reviewed drafts of the article, and approved the final draft.
- Kazuyo Nakaoka conceived and designed the experiments, authored or reviewed drafts of the article, and approved the final draft.
- Yasuhiro Higashi conceived and designed the experiments, authored or reviewed drafts of the article, and approved the final draft.
- Yoshimi Yuri conceived and designed the experiments, authored or reviewed drafts of the article, and approved the final draft.
- Hiroko Hashimoto conceived and designed the experiments, authored or reviewed drafts of the article, and approved the final draft.
- Shinichi Takabatake conceived and designed the experiments, authored or reviewed drafts of the article, and approved the final draft.

### Human Ethics

The following information was supplied relating to ethical approvals (*i.e.*, approving body and any reference numbers):

The Graduate School of Comprehensive Rehabilitation, Osaka Prefecture University's research ethics committee.

### Data Availability

The raw data is available in the Supplemental Files.

## Supplemental Information

Supplemental information for this article can be found online at http://dx.doi.org/10.7717/peerj.17730#supplemental-information.

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
