# Peer review of "Examination of reliability and validity of the Self-Assessment Burden Scale-Motor for community-dwelling older adults in Japan: a validation study"

_PeerJ, doi:10.7717/peerj.17730_

## Round 0.1 · original submission · Minor Revisions

Thanks for submitting this work. The independent reviewers have both indicated minor revisions, so if you can please make those changes and send us within the reasonable time, I believe we willl be able to publish it. Congratulations on getting the work done well and thanks in advance.

Reviewer 1 ·

Basic reporting

I only have a minor comment: the space before parenthesis is often missing (e.g., line 101: family caregivers(Kaneda et al. ..)

Experimental design

8.7% of the participants had missing data. Can you please explain why the data are missing?

Validity of the findings

The stats reported in the results section seem to be incomplete. For example, What is the statistical significance (p value) for the correlation? Can you please report the statistics for correlation with p value as well as sample size in the result, similar to the following example? Otherwise, it would be hard to understand the validity.

"The analysis revealed a strong positive correlation between variable X and variable Y, rs(166) = 0.91, p < 0.001"

Similar with the reporting of the weighted kappa.

·

Basic reporting

Thank you for your valuable work these are some recommendations for
Background
1- It is better to replace “community-dwelling older” with a better phrase.
2- What do you mean by “physical activity interventions” please explain it.
3- There are many old references from 2001-2002-2008.
4- some of the words and phrases are not common like “incurs “
5- The background does not shoes that what “home-visit rehabilitation” means please explain it.
6- Please explain why the Caregiver rating is important.
7- There is no information about the Self-Assessment Burden Scale-Motor (SAB-M) in the background.
8- The background is not very clear.

Material and method
1- The first inclusion criteria was “staying at home for at least 3 months” but you did not mention it in the background.
2- In Lines 136-141 the author tried to introduce the Self-Assessment Burden Scale-Motor (SAB-M) but it was not enough. It is not clear who made it or when it was made if its reliability and validity were tested or not. If factor analysis done or not?
3- Line 170 needs a grammar check
4- Line 250 “8.7% of the data were missing” What does it mean? What was your solution
5- Line 251 needs to be edited “whereas ours had a lower rate”
6- Lines 274 – 277 should be omitted or moved to the background.
7- Lines 287 -291 should be omitted or moved it to the background.
8- Lines 294 -298 should be moved to method and material.
9- Line 313 is the limitation of your study

Experimental design

.

Validity of the findings

your sincerely

Additional comments

no comment'

---

## Round 0.2 · accepted · Accept

This is to confirm that the authors have addressed all of the reviewers' comments, and the manuscript is now ready for publication.

Reviewer 1 ·

Basic reporting

I have no comment. Thank you for the revisions!

Experimental design

I have no comment. Thank you for the revisions!

Validity of the findings

I have no comment. Thank you for the revisions!